# Changes in Cardiorespiratory Fitness during a Season in Elite Female Soccer, Basketball, and Handball Players

Evangelia Papaevangelou [1], Zacharoula Papadopoulou [1,2,3], Yiannis Michailidis [2,*], Athanasios Mandroukas [2], Pantelis T. Nikolaidis [4], Nikolaos V. Margaritelis [3] and Thomas Metaxas [2]

[1] School of Physical Education and Sports Science, Department of Competitive Sports, Division of Team Handball, Aristotle University of Thessaloniki, 57001 Thessaloniki, Greece; papaevangelou@yahoo.com (E.P.); pzaxaro@phed.auth.gr (Z.P.)

[2] Laboratory of Evaluation of Human Biological Performance, Department of Physical Education and Sports Science, Aristotle University of Thessaloniki, New Buildings of Laboratories, University Campus of Thermi, 57001 Thessaloniki, Greece; amandrou@phed.auth.gr (A.M.); tommet@phed.auth.gr (T.M.)

[3] Laboratory of Exercise Physiology & Biochemistry, Department of Physical Education and Sports Science at Serres, Aristotle University of Thessaloniki, 57001 Thessaloniki, Greece; nvmargar@phed-sr.auth.gr

[4] School of Health and Caring Sciences, University of West Attica, 12243 Athens, Greece; pademil@hotmail.com

* Correspondence: ioannimd@phed.auth.gr; Tel.: +30-23-1099-2233

**Abstract:** The purpose of the study was to examine and detect the changes in cardiorespiratory fitness of female soccer, basketball, and handball players during an annual training cycle. In the study, 19 soccer players ($23.2 \pm 4.3$ years), 26 basketball players ($21.1 \pm 5.4$ years), and 26 handball players ($21.1 \pm 4.2$ years) participated. All participants underwent anthropometric measurements and a laboratory maximal oxygen uptake test ($VO_{2max}$) at the beginning of the preparation training period, in the middle of the competitive season, and at the end of it. $VO_{2max}$ showed significant changes in all three team sports from the initial measurement to the final, with a significant value increase in the middle of the competitive season and a small drop at the end (soccer $52.1 \pm 5.6$ vs. $59.6 \pm 6.3$ vs. $53.5 \pm 5.4$, basketball $44.6 \pm 5.8$ vs. $50.3 \pm 8.5$ vs. $45.9 \pm 6.4$, and handball $47.9 \pm 4.8$ vs. $49.1 \pm 5.3$ vs. $46.7 \pm 4.9$ mL/kg/min) ($p < 0.05$), respectively. In conclusion, the cardiorespiratory fitness of female soccer, basketball, and handball players was significantly improved in the middle of the competitive season, probably due to the development of physical capacity. The decrease in cardiorespiratory fitness at the end of the competitive season, which was observed in all three team sport athletes, possibly occurred due to the differentiation in the training program.

**Keywords:** soccer; basketball; handball; anthropometric measurements; cardiorespiratory fitness





## 1. Introduction

Soccer, basketball, and handball are intermittent sports that include prolonged phases of low, moderate, and high intensity [1–3]. Previous studies have shown that female soccer athletes cover approximately 9.2 km [4], basketball athletes cover 5.2 km [5], while handball athletes cover roughly 4 km [6]. Sports of this type involve both aerobic and anaerobic energy production mechanisms [7], and although the essential actions in each sport are characterized by high intensity, high aerobic capacity can improve the ability to repeat sprints and to recover faster between them [8], helping to maintain performance during a match [9]. In addition, female athletes with better aerobic capacity have rapid replenishment of phosphocreatin stores [10], which is necessary for rapid actions. The contribution of aerobic metabolism to the energy requirements of each sport is shown by aerobic indicators that have been measured during sports. Thus, for soccer, it is reported that the aerobic mechanism provides approximately 90% of total energy, with exercise intensity corresponding to 70% of $VO_{2max}$ [11]. The corresponding intensity for basketball is 65% of $VO_{2max}$ [12], while for handball, it has been reported that the intensity of the

match corresponds to 80–90% of the maximum heart rate [13]. From the above references, the importance of aerobic mechanisms in these three sports becomes obvious.

Aerobic capacity was assessed with the measure of maximum oxygen uptake ($VO_{2max}$), which referred to the maximal ability of the body to use oxygen during maximal effort. Elite female soccer players presented values of $VO_{2max}$ varying from 49 to 58 mL/kg/min [14]. A similar large variation occurs in the values reported for female basketball athletes (from 44 to 54 mL/kg/min) [15] and female handball athletes (from 42 to 58 mL/kg/min) [16]. The large variation reported in the studies is likely due to differences between participants. Although we refer to elite female athletes, the level of competition of female athletes can vary significantly between countries. More specifically, the first division of a sport may be professional in one country and amateur in another. We realize that these differences can affect the level of female athletes as they are related to the training process (training volume, training quality, etc.). Therefore, studies may report that they take place in elite athletes, but the differences from country to country are large.

It is known that aerobic capacity can be affected by the training program applied during the annual training cycle as the characteristics of the load vary [17]. The three sports in the study have extended racing periods of more than 9 months. Coaches should maintain the high performance of their athletes for the entirety of time by differentiating the characteristics of the load [18].

The number of studies on the annual change in $VO_{2max}$ in female soccer is limited [17,19,20]. Additionally, of these, only two [17,20] have made more than two measurements during the year in the same athletes. These studies were conducted in Division I and II, respectively, of the National Collegiate Athletic Association. They made multiple measurements, but none of them were taken during the season. In the study conducted in Division I, they observed that aerobic capacity decreased in the pre-season period when training included indirect exercises [20]. In contrast, in the study in Division II, no differences in aerobic capacity were observed [17]. As far as basketball and handball are concerned, studies are minimal [21–23]. In a recent study [24] of U18 teams competing in a national division, three measurements were performed during the year and field tests were applied. The results showed that aerobic capacity improves throughout the year as a result of the training process. In one study in handball [25], they applied two different periodization models for two consecutive years and compared their effect on fitness indicators. The results showed that both models improved aerobic capacity at the end of each racing year. Aerobic capacity was measured by laboratory testing, but no measurements were made during the year. In contrast, in another study [22] of elite female handball players who applied field tests to assess aerobic capacity four times during the year, they observed no significant changes. Finding changes in aerobic fitness during the annual training cycle will help coaches identify the needs of their athletes and design appropriate training programs.

Thus, the purpose of this study was to examine and investigate changes in the cardiorespiratory fitness of female soccer, basketball, and handball athletes during the annual training cycle. In addition, we aimed to compare the results between the measurements in the three different sports before the start of the preparation, in the middle of the season, and at the end of the season.

## 2. Material and Methods

### 2.1. Study Design

Before the beginning of the study, all players provided written consent, and they were fully informed about the procedures of the experiment. The study was performed in accordance with the local University Ethics Committee guidelines and with the ethical standards of sports medicine research (protocol code 7102/2014). All subjects completed a questionnaire that included their relevant medical history.

Anthropometric measurements and aerobic capacity measurements were performed 3 times over the annual training cycle of each sport. Aerobic capacity changes were analyzed by comparing the test results from one stage of the season to the next. Additionally, the

results of each test were compared between sports. All the measurements of each test day were performed during a single visit to the laboratory. Participants were informed before each testing block to avoid intense exercise and not receive any kind of medication.

All athletes were members of the First Division for each sport, respectively. The minimum training experience of all athletes was three years, and the minimum training frequency was five training sessions per week and competition in one game per week throughout the season. Of their training, two were specific to fitness and the rest towards technical, tactical elements. However, in each workout, there were some elements of fitness. According to the classification of McKay et al. (2021) [26], the athletes of all three sports belong to tier 3: highly trained/national level. The first testing (1st) took place at the beginning of the preparation training period (in the first week), the second one (2nd) in the middle of the competitive season, and the third one (3rd) at the end for each sport, respectively. The athletes' championships ended at the end of May, but training continued for another month. A month followed, where during the first two weeks, the athletes abstained from organized sports activities, and then the other two weeks implemented personalized programs aimed at basic endurance and strength. Then, the preparation of the female athletes began. The study design is presented in Figure 1.

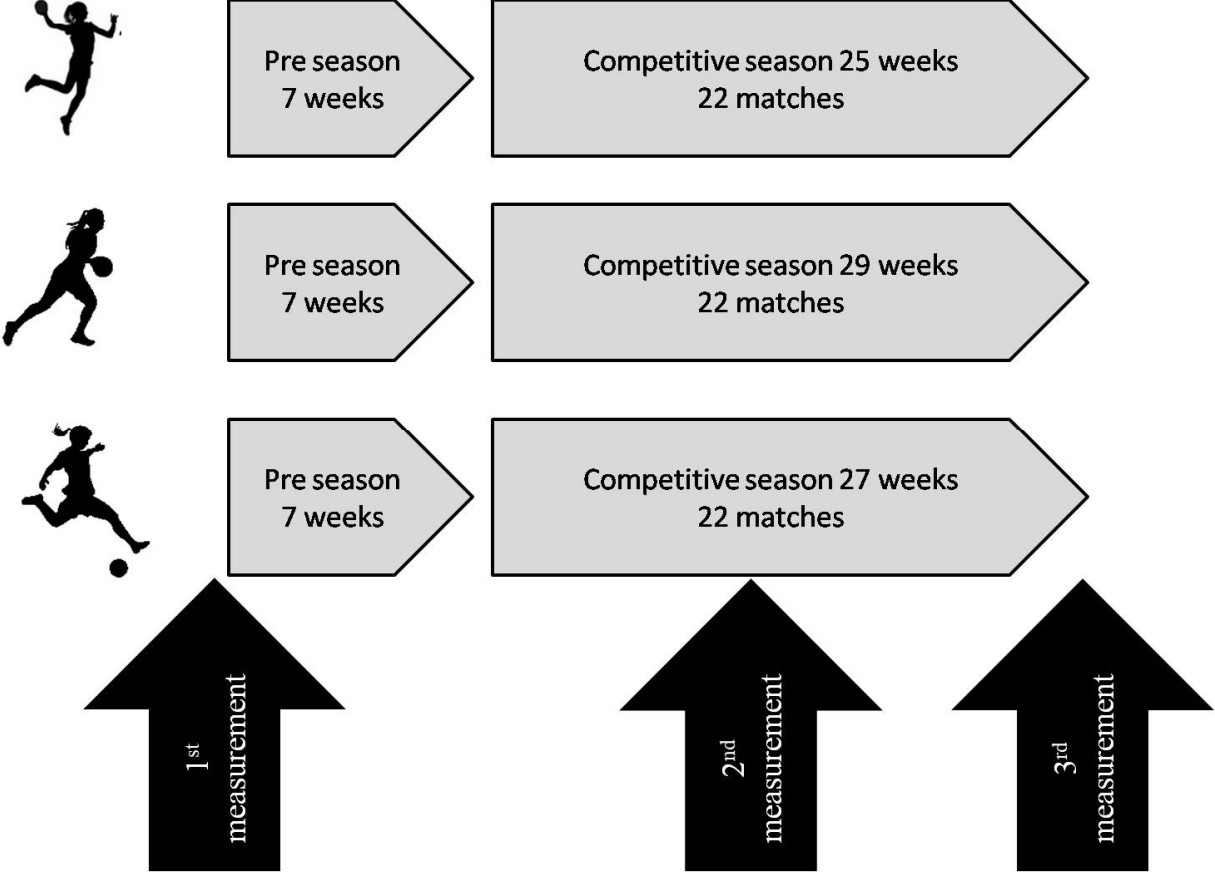

**Figure 1.** Study design.

## 2.2. Subjects

The study comprised 71 female team sport athletes, 19 soccer players (mean ± SD: aged 23.2 ± 4.3 years and mean training experience 8.2 ± 3.6 years), 26 basketball players (mean ± SD: aged 21.1 ± 5.4 years and mean training experience 9.0 ± 4.7 years), and 26 handball players (mean ± SD: aged 21.1 ± 4.2 years and mean training experience 10.9 ± 4.7 years).

Training Program

Table 1 below shows the main objectives of the training sessions during the season. The first six weeks after the first measurement constituted the preparation period, and in the first four of them, the elements of fitness dominated as training goals.

**Table 1.** Basic training characteristics during the competitive season.

| | Monday | Tuesday | Wednesday | Thursday | Friday | Saturday | Sunday |
|---|---|---|---|---|---|---|---|
| Soccer | Recovery | Aerobic training<br>Technical–Tactical training | Power<br>Speed<br>Strength | Specific endurance<br>Technical–Tactical training | Technical–Tactical training | Day off | Game |
| Basketball | Recovery | Aerobic training<br>Technical–Tactical training | Strength<br>Technical–Tactical training | Power<br>Technical–Tactical training | Technical–Tactical training | Day off | Game |
| Handball | Recovery | Aerobic training<br>Technical–Tactical training | Technical–Tactical training<br>Strength | Power<br>Speed<br>Technical–Tactical training | Technical–Tactical training | Day off | Game |

### 2.3. Anthropometric Measurements

All participants underwent anthropometric examination, including stature and body mass measurements using an electronic digital scale (Seca 220e, Hamburg, Germany) and also body-fat assessment using skinfold measurements (four-fold method): biceps ($S_1$), triceps ($S_2$), suprailiac ($S_3$), and subscapular ($S_4$) by specific calliper (DrLange, Santa Cruz, California) (Table 2). Estimation of body density was calculated according to standard equations for females over 16 years old [27], and the percentage of body fat was estimated by the equation of Siri (1956) [28]. Additionally, the lean body mass (LBM) was calculated from the body mass and body fat measurements.

**Table 2.** Anthropometric characteristics of participants by sport at baseline (Mean $\pm$ SD).

| | Soccer | Basketball | Handball |
|---|---|---|---|
| Stature (cm) | 164.9 $\pm$ 6.0 *** | 175.4 $\pm$ 5.7 $^{\$\$\$}$ | 167.3 $\pm$ 4.7 |
| Body mass (kg) | 58.7 $\pm$ 6.1 *** | 71.6 $\pm$ 13.7 $^{\$\$}$ | 63.1 $\pm$ 6.5 |
| Biceps (mm) | 4.9 $\pm$ 2.2 *### | 7.5 $\pm$ 3.8 | 8.5 $\pm$ 3.1 |
| Triceps (mm) | 9.4 $\pm$ 4.9 ***### | 16.3 $\pm$ 5.6 | 17.8 $\pm$ 7.0 |
| Suprailiac (mm) | 7.8 $\pm$ 2.9 ***## | 13.3 $\pm$ 4.8 | 11.5 $\pm$ 3.7 |
| Subscapular (mm) | 7.8 $\pm$ 1.4 *# | 9.8 $\pm$ 3.3 | 9.9 $\pm$ 2.3 |
| Lean body mass (kg) | 46.7 $\pm$ 3.7 ** | 52.0 $\pm$ 7.6 $^{\$\$\$}$ | 46.2 $\pm$ 3.2 |
| Body fat (%) | 20.2 $\pm$ 3.6 ***### | 26.4 $\pm$ 4.4 | 26.6 $\pm$ 4.7 |

Soccer vs. Basketball: *: $p < 0.05$; **: $p < 0.01$; ***: $p < 0.001$; Soccer vs. Handball: #: $p < 0.05$; ##: $p < 0.01$; ###: $p < 0.001$; Basketball vs. Handball: $\$\$$: $p < 0.01$; $\$\$\$$: $p < 0.001$.

### 2.4. Cardiorespiratory Endurance

The assessment of $VO_{2max}$ was performed on a motorized treadmill (HP Cosmos, Pulsar, Nussdorf-Traustein, Germany) using a continuous exercise testing protocol. The initial grade and speed were set at 0% at 8 km/h. Every two minutes, the speed increased by 2 km/h up to a speed of 12 km/h. Then, from a speed of 14 km/h, the gradient was set at 2%, and the speed increased every 1 min until exhaustion. The oxygen uptake, determined by means of absolute (mL/min) and relative values adjusted to body weight (mL/kg/min), as well as the cardiorespiratory indices, were measured via an ergospirometric device based

on a breath-by-breath automated pulmonary/metabolic gas exchange system (Oxycon Pro-Jaeger, Würzburg, Germany) using a tight face mask specially designed for children. The HR was recorded continuously using a Polar HR monitor (Polar Electro, Oy, Kempele, Finland) connected to the ergospirometric device. $VO_{2max}$ was assumed when three of the four following criteria were met: (a) the HR during the last minute exceeded 95% of the expected maximal HR predicted 220-age; (b) a respiratory exchange ratio (RER) $(VCO_2/VO_2)$ at or higher than 1.1 was reached; (c) $VO_2$ reached a plateau and/or signs of subjective exhaustion were present and the subject was unable to continue running, despite verbal encouragement [29]. Blood samples were obtained from the hand-warmed fingertip and the concentration of blood lactate was determined in the 5th minute of recovery using a lactate photometer analyzer (Accusport, Boegringer Manheim, Germany). The rating of perceived exertion (RPE) was obtained during maximal exercise by use of a general Borg 6–20 scale [30].

During the test, the following additional cardiorespiratory parameters were determined: exercise duration; the maximal pulmonary ventilation ($VE_{max}$), the heart rate ($HR_{AT}$), and the $VO_2$ at the anaerobic threshold; the speed of anaerobic threshold ($U_{AT}$), the maximal heart rate ($HR_{max}$), and the respiratory exchange ratio (RER).

Blood Pressure (BP)

Arterial BP (systolic arterial pressure, SAP, and diastolic arterial pressure, DAP) and resting heart rate (HR) were measured in a resting state, with the participants in the supine position with their legs uncrossed and the middle of the cuff at the level of the right atrium of their heart. Measurements were performed in a quiet room under standarized conditions between 9:00 and 10:00 h to avoid diurnal variations [31]. The participants rested for at least 5 min in order to acclimate to the new environment and to allow their blood pressure to normalize. An automatic BP monitor (M7, Omron, Vernon Hills, IL, USA) was used with a cuff adjusted to the arm size as appropriate. Resting HR was measured simultaneously by the automatic PB monitor.

*2.5. Statistics*

Data are presented as mean ±SD. A Kolmogorov–Smirnov test was used to test the normality, while Levenne's test was used to assess the equality of variances, respectively. Two-way ANOVA with repeated measures was used to detect the differences between groups as well as the differences between the testing in time for every group, respectively. Where appropriate, Bonferroni post hoc tests were used to determine which points were significantly different. Partial eta squared values were also reported, with these classified as small (0.01–0.059), moderate (0.06–0.137), and large (>0.138) [32]. The statistical analysis was performed via H/Υ PASW (Predictive Analytics Software) Statistics for Windows Version 18.0, 2009 (SPSS Inc. An IBM Company, Chicago, IL, USA). The level of statistical significance was set at $p < 0.05$.

**3. Results**

Changes in anthropometric characteristics are presented in Table 3.

**Table 3.** Anthropometric characteristics by time (Mean ± SD).

| | Measurements | | | Statistics | | |
|---|---|---|---|---|---|---|
| | 1st | 2nd | 3rd | F | *p* | $\eta^2$ |
| | | Soccer | | | | |
| Body mass (kg) | 58.7 ± 6.1 | 57.9 ± 6.1 | 57.1 ± 5.8 | 0.872 | 0.424 | 0.031 |
| Lean body mass (kg) | 46.7 ± 3.7 | 46.5 ± 45.7 | 46.2 ± 3.8 | 0.351 | 0.706 | 0.045 |
| Body fat (%) | 20.2 ± 3.6 | 19.7 ± 3.4 | 19.1 ± 3.2 | 1.272 | 0.289 | 0.013 |

**Table 3.** *Cont.*

|  | Measurements | | | Statistics | | |
|---|---|---|---|---|---|---|
|  | 1st | 2nd | 3rd | F | *p* | $\eta^2$ |
|  | Basketball | | | | | |
| Body mass (kg) | 71.6 ± 13.7 | 70.8 ± 13.3 | 69.7 ± 12.8 | 0.644 | 0.528 | 0.017 |
| Lean body mass (kg) | 52.0 ± 7.6 | 52.7 ± 7.7 | 52.1 ± 7.3 | 0.209 | 0.812 | 0.006 |
| Body fat (%) | 26.4 ± 4.4 | 25.6 ± 4.7 | 25.2 ± 4.3 | 1.782 | 0.175 | 0.045 |
|  | Handball | | | | | |
| Body mass (kg) | 63.1 ± 6.5 | 62.3 ± 6.4 | 61.9 ± 6.1 | 2.094 | 0.130 | 0.053 |
| Lean body mass (kg) | 46.2 ± 3.2 | 46.4 ± 3.4 | 45.0 ± 3.3 | 0.887 | 0.416 | 0.023 |
| Body fat (%) | 26.4 ± 4.7 | 25.5 ± 4.0 | 25.1 ± 3.8 | 2.073 | 0.133 | 0.052 |

In soccer players, the $HR_{AT}$, UAT, and absolute $VO_{2max}$ increased during the second measurement and differed with the first and third measurements (F = 30.302, $\eta^2$ = 0.529, $p$ = 0.007—F = 8.907, $\eta^2$ = 0.169, $p < 0.001$—F = 7.177, $\eta^2$ = 0.210, $p < 0.001$, respectively). The relative $VO_{2max}$ differed between the measurements (F = 8.917, $\eta^2$ = 0.248, $p < 0.001$). No other differences were observed in any other variable (Table 4).

**Table 4.** Cardiorespiratory fitness test by time (Mean ± SD).

|  | Measurements | | |
|---|---|---|---|
|  | 1st | 2nd | 3rd |
|  | Soccer | | |
| $HR_{rest}$ (b/min) | 67.2 ± 6.9 | 66.8 ± 6.9 | 67.7 ± 7.4 |
| SAP (mmHg) | 116.3 ± 6.5 | 114.9 ± 6.4 | 114.8 ± 8.6 |
| DAP (mmHg) | 58.6 ± 5.9 | 57.8 ± 6.8 | 58.1 ± 7.1 |
| Exercise time (min) | 7.34 ± 0.62 | 7.10 ± 0.75 | 7.17 ± 0.50 |
| $HR_{AT}$ (b/min) | 147.7 ± 6.5 *** | 163.3 ± 8.8 $^{\$\$\$}$ | 147.8 ± 5.6 |
| $U_{AT}$ (km/h) | 11.8 ± 1.2 *** | 13.2 ± 0.8 $^{\$}$ | 12.3 ± 1.1 |
| $HR_{max}$ (b/min) | 197.3 ± 3.0 | 196.8 ± 4.3 | 194.9 ± 3.1 |
| $VE_{max}$ (L/min) | 98.6 ± 10.9 | 101.1 ± 9.4 | 97.4 ± 13.4 |
| $U_{max}$ (km/h) | 14.9 ± 1.2 | 14.4 ± 1.5 | 14.5 ± 0.9 |
| Absolute $VO_{2max}$ (mL/min) | 3046.4 ± 356.9 *** | 3412.3 ± 432.0 $^{\$\$\$}$ | 2989.8 ± 322.1 |
| Relative $VO_{2max}$ (mL/kg/min) | 52.1 ± 5.6 ***# | 59.6 ± 6.3 $^{\$\$\$}$ | 53.5 ± 5.4 |
| RER | 1.11 ± 0.04 | 1.12 ± 0.04 | 1.13 ± 0.26 |
| Blood lactate (mmol/L) | 9.0 ± 1.5 | 9.3 ± 1.3 | 8.8 ± 1.7 |
|  | Basketball | | |
| $HR_{rest}$ (b/min) | 66.8 ± 9.3 *** | 64.1 ± 8.5 $^{\$\$}$ | 70.3 ± 11.8 |
| SAP (mmHg) | 127.5 ± 11.7 ***### | 123.4 ± 8.7 $^{\$\$\$}$ | 116.7 ± 6.9 |
| DAP (mmHg) | 63.6 ± 5.1 ***### | 59.9 ± 4.4 $^{\$\$\$}$ | 64.5 ± 7.8 |
| Exercise time (min) | 7.73 ± 0.65 **### | 7.30 ± 0.80 | 7.25 ± 0.78 |
| $HR_{AT}$ (b/min) | 144.2 ± 7.1 ** | 152.0 ± 11.6 $^{\$}$ | 147.3 ± 7.2 |
| $U_{AT}$ (km/h) | 11.8 ± 1.2 ***# | 13.0 ± 1.1 | 12.4 ± 1.2 |
| $HR_{max}$ (b/min) | 198.1 ± 4.5 | 197.5 ± 5.0 | 196.6 ± 5.2 |
| $VE_{max}$ (L/min) | 101.6 ± 14.5 # | 105.8 ± 16.5 | 106.8 ± 17.1 |
| $U_{max}$ (km/h) | 15.9 ± 1.5 ### | 15.3 ± 1.6 $^{\$}$ | 14.6 ± 1.5 |
| Absolute $VO_{2max}$ (mL/min) | 3148.8 ± 478.1 ***# | 3502.7 ± 824.4 $^{\$\$\$}$ | 3045.2 ± 436.5 |
| Relative $VO_{2max}$ (mL/kg/min) | 44.6 ± 5.8 ***## | 50.3 ± 8.5 $^{\$\$\$}$ | 45.9 ± 6.4 |
| RER | 1.13 ± 0.05 | 1.12 ± 0.06 | 1.11 ± 0.07 |
| Blood lactate (mmol/L) | 8.3 ± 1.1 | 8.9 ± 1.1 | 8.4 ± 1.7 |

**Table 4.** *Cont.*

| | Measurements | | |
| | 1st | 2nd | 3rd |
|---|---|---|---|
| | | Handball | |
| $HR_{rest}$ (b/min) | 70.1 ± 8.1 ** | 68.2 ± 9.6 $^{\$\$}$ | 74.0 ± 10.6 |
| SAP (mmHg) | 115.3 ± 7.3 ** | 112.7 ± 8.6 | 114.4 ± 8.1 |
| DAP (mmHg) | 67.5 ± 6.5 ** | 66.0 ± 7.8 | 67.5 ± 8.0 |
| Exercise time (min) | 7.90 ± 0.58 | 7.74 ± 0.72 | 7.68 ± 0.77 |
| $HR_{AT}$ (b/min) | 146.7 ± 4.8 * | 152.8 ± 10.9 $^{\$\$}$ | 147.1 ± 6.9 |
| $U_{AT}$ (km/h) | 12.1 ± 1.1 | 12.6 ± 0.9 | 12.6 ± 1.0 |
| $HR_{max}$ (b/min) | 197.8 ± 3.7 # | 197.7 ± 4.0 $^{\$}$ | 195.2 ± 4.4 |
| $VE_{max}$ (L/min) | 99.7 ± 8.8 | 99.2 ± 9.9 | 97.5 ± 8.3 |
| $U_{max}$ (km/h) | 16.2 ± 1.2 | 15.8 ± 1.3 | 15.6 ± 1.6 |
| Absolute $VO_{2max}$ (mL/min) | 3017.0 ± 425.0 ### | 3027.2 ± 390.8 $^{\$}$ | 2819.9 ± 339.8 |
| Relative $VO_{2max}$ (mL/kg/min) | 47.9 ± 4.8 # | 49.1 ± 5.3 $^{\$}$ | 46.7 ± 4.9 |
| RER | 1.14 ± 0.05 ### | 1.12 ± 0.04 $^{\$\$\$}$ | 1.07 ± 0.07 |
| Blood lactate (mmol/L) | 9.9 ± 1.5 ## | 9.8 ± 1.2 $^{\$\$}$ | 8.7 ± 1.6 |

SAP—systolic artery pressure; DAP—diastolic artery pressure; $HR_{AT}$—heart rate anaerobic threshold; $U_{AT}$—velocity$_{anaerobic\ threshold}$; $HR_{max}$—heart rate maximal; $VE_{max}$—maximal exercise ventilation; $U_{max}$—maximal velocity at test; RER—respiratory exchange ratio. First measurement vs. second measurement: *: $p < 0.05$; **: $p < 0.01$; ***: $p < 0.001$; first measurement vs. third measurement: #: $p < 0.05$; ##: $p < 0.01$; ###: $p < 0.001$; second measurement vs. third measurement: $: $p < 0.05$; $$: $p < 0.01$; $$$: $p < 0.001$.

$HR_{AT}$ of basketball players differed between the first and second measurements (F = 5.050, $\eta^2 = 0.119$, $p < 0.001$) and between the second and third measurements ($p = 0.007$). Differences in $U_{AT}$ were observed between the first measurements compared to the other two measurements (F = 7.617, $\eta^2 = 0.169$, 2nd: $p < 0.001$, 3rd: $p = 0.018$). Similar differences were observed for the time of exercise testing (F = 3.289, $\eta^2 = 0.081$, 1st vs. 2nd $p = 0.008$, 1st vs. 3rd $p < 0.001$). $U_{max}$ decreased during the season (F = 5.119, $\eta^2 = 0.120$, 1st vs. 3rd $p < 0.001$, 2nd vs. 3rd $p = 0.006$). The absolute $VO_{2max}$ (F = 4.085, $\eta^2 = 0.098$, $p = 0.001$) and relative $VO_{2max}$ (F = 4.828, $\eta^2 = 0.114$, $p < 0.001$) differed between time points. No other changes were observed for other variables (Table 4).

Handball players showed differences in $HR_{AT}$ between the first and second measurements (F4.686, $\eta^2 = 0.119$, $p = 0.012$) and between the second and third measurements ($p < 0.001$). Additionally, $HR_{max}$ decreased at the third measurement (F = 3.419, $\eta^2 = 0.084$, $p < 0.05$). The absolute $VO_{2max}$ at the third measurement decreased and differed compared to the first (F = 8.425, $\eta^2 = 0.038$, $p < 0.001$) and second measurements ($p < 0.05$). Similar results for relative $VO_{2max}$ between the first and third (F = 5.632, $\eta^2 = 0.016$, $p = 0.042$) and second and third measurements ($p = 0.039$). Blood lactate decreased during the season (F = 5.174, $\eta^2 = 0.121$, 1st vs. 3rd $p = 0.008$ and 2nd vs. 3rd $p = 0.008$). Similar changes were observed for RER between the third measurement and the first and second measurements (F = 9.738, $\eta^2 = 0.206$, $p < 0.001$). No other changes were observed (Table 4).

Between the sports, no differences were observed for $HR_{rest}$ (F = 1.147, $p = 0.325$, $\eta^2 = 0.346$). Exercise time to exhaustion was longer for soccer players in comparison with handball players during the first (F = 4.474, $p = 0.013$, $\eta^2 = 0.245$) and second measurements (F = 4.351, $p = 0.019$, $\eta^2 = 0.245$). The $HR_{AT}$ differed in the second measurement between soccer players, basketball players, and handball players (F = 7.396, $\eta^2 = 0.395$ 163.3 ± 8.8 vs. 152.0 ± 11.6, $p = 0.002$, and 163.3 ± 8.8 vs. 152.8 ± 10.9, $p < 0.001$). The absolute values of $VO_{2max}$ were different at the second measurement between basketball and handball players (F = 4.561, 3502.7 ± 824.4 vs. 3027.2 ± 390.8, $p = 0.016$, $\eta^2 = 0.408$). Relative values of $VO_{2max}$ (mL/kg/min) were higher for soccer players in comparison with basketball (F = 10.619, $\eta^2 = 0.521$ 52.1 ± 5.6 vs. 44.6 ± 5.8, $p < 0.001$) and handball players (52.1 ± 5.6 vs. 47.9 ± 4.8, $p = 0.033$). The differences between the second (F = 14.583, $p < 0.001$, for both sports) and third measurements (F = 11.695, $p < 0.001$ for both sports) were similar. Blood lactate differed at the first measurement between basketball and handball players

(F = 9.094, 8.3 $\pm$ 1.1 vs. 9.9 $\pm$ 1.5, $p < 0.001$, $\eta^2 = 0.187$). No other differences were observed between the sports.

## 4. Discussion

The purpose of this study was to examine and investigate changes in the cardiorespiratory performance of female soccer, basketball, and handball athletes during the annual training cycle. In addition, we aimed to compare the results between the measurements in the three different team sports before the start of preparation, in the middle, and at the end of the season. The results of the study showed that in all three sports, the maximum oxygen uptake increased in the middle of the season, while at the end of the season, it decreased to initial levels and handball even lower.

In the anthropometric characteristics, the differences observed in the first measurement continued to appear in the other two measures between sports. The differences between the three measurements showed no significant differences. However, in all sports, there was a slight decrease in body mass, lean body mass, and body fat percentage. This decline was progressive between the three measurements. Body weight affects relative $VO_{2max}$. An increase in relative $VO_{2max}$ was observed in the study, but it cannot be attributed to a decrease in body mass, as in the third measurement, while there was an additional drop in body weight, there was no increase in relative $VO_{2max}$. On the contrary, there was a decrease in initial levels.

$VO_{2max}$ (absolute and relative) was higher in soccer players from the start of the preparation until the end of the season compared to the other two team sports. As mentioned above, aerobic capacity plays an important role in the performance of female soccer players, perhaps greater than the other two sports. This leads coaches and athletes to place greater emphasis on this ability. Additionally, the field in which the female soccer athletes train is much larger compared to the fields of the other two sports. As a result, female soccer players cover longer distances during training sessions, resulting in higher maximum oxygen uptake.

The evaluation of cardiorespiratory performance found that the relative $VO_{2max}$ values of soccer athletes in this study were higher than those of Australian college athletes, Turkish athletes, and England international athletes [33], while they were similar to those of the Japanese national soccer team and England's high-level athletes [34]. The values were lower than those mentioned for Danish female athletes [34]. $VO_{2max}$ (absolute and relative) of soccer athletes in this study showed changes between measurements, increasing significantly in the middle of the season and falling at the end of the season. The relative $VO_{2max}$ of basketball athletes was similar to those reported in previous studies for high-level basketball athletes [5,12] and showed an increase in the middle of the season with a fall that followed at the end of it. In handball athletes, as in the female athletes of the other two team sports, an increase in $VO_{2max}$ was observed in the second measurement with a drop in the third measurement to starting levels. The relative $VO_{2max}$ of athletes in this study was similar to those reported in previous studies on handball athletes [16] and lower than some previous studies [35]. The increase in relative $VO_{2max}$ observed in the middle of the season and the drop at the end is also in line with the research of Caldwell and Peters (2009) [36], who observed a drop in fatigue of the soccer athletes in their study, as well as the burden of the second half of the season. The values of handball athletes were higher compared to basketball athletes and lower than soccer athletes, who showed the highest values in all three measurements during their annual training cycle compared to the other two team sports. In the exercise time until exhaustion, there was no difference between the three measurements in the soccer and handball athletes. Basketball athletes showed a difference between the three measurements. Differences between the three sports for the exercise time to exhaustion were observed only between the athletes of soccer and handball in the first and second measurements. $HR_{max}$, RER, and blood lactate at the end of the test showed significant differences between measurements only in handball athletes, while $HR_{max}$ found no differences between the three sports in any measurement. Finally,

in blood lactate, a significant difference was observed between basketball and handball athletes only in the first measurement.

The evaluation of cardiorespiratory performance in team sports in laboratory conditions helps significantly in understanding the physiology of these sports. In addition, the results of the tests provide important information on the physiological requirements of the sport and the abilities of the athletes while helping in the coaching of athletes with training programs for their development and improvement [15]. More specifically, the tests can obtain information about the intensity at which female athletes need to exercise their aerobic capacity. In this way, exercise becomes more specialized, resulting in greater adjustments. It is also important for high performance in matches to enhance the individual skills of athletes and to identify and correct individual weaknesses in training to improve the overall performance of the team [37]. The training design must be conducted in such a way that it is individualized to each athlete, to improve and evolve. In team sports, the training takes place mainly in all athletes (team training) so that the athlete does not follow a specialized training program based on her individual and physiological characteristics. This probably leads, on the one hand, female athletes who excel in physical fitness not to accept the appropriate training stimulus that will also help to improve them [38], and on the other hand, female athletes who fall short in physical abilities are forced to work more intensively to reach the same level as other female athletes resulting in increased fatigue. This situation leads to performance reduction and increased risk of injury [39]. However, at a high level in recent years, tests have been applied so that female athletes can exercise at intensities that correspond to their individual characteristics. Additionally, the development of technology (e.g., global positioning systems, GPS) allows the monitoring of training intensities and training load, making training more efficient and protecting female athletes from fatigue and possible injuries.

The study also has some limitations. Initially, this was a descriptive study of three team sports where the sample was limited, and observations could not be generalized. Although we knew the basic characteristics of training (intensity, volume, etc.) in each sport, they were not recorded in detail in each microcycle.

The results of the study showed that cardiorespiratory performance improved significantly from baseline compared to mid-season measurement in all three sports. However, this increase was not maintained, resulting in a decrease in $VO_{2max}$ to baseline levels at the end of the season.

In conclusion, it was found that the physiological characteristics of soccer, basketball, and handball athletes showed a significant improvement in the middle of the season, which can be attributed to the development of mainly physical abilities in the preparation period, the start of racing obligations, and the practical application of laboratory data from athletes' measurements. On the contrary, the significant drop in cardiorespiratory performance at the end of the season observed in the female athletes of all three sports is likely due to the differentiation of the training schedule and the racing burden, as well as the probable change in motivation of the athletes who participated in the study.

**Author Contributions:** Conceptualization, E.P., Z.P. and T.M.; methodology, E.P., A.M., Y.M. and P.T.N.; software, E.P., Z.P., A.M. and N.V.M.; validation, E.P., Z.P. and T.M.; formal analysis, E.P., A.M., Y.M. and P.T.N.; investigation, E.P., Z.P. and T.M.; resources, E.P., Z.P. and T.M.; data curation, E.P., A.M., Y.M. and T.M.; writing—original draft preparation, E.P., Z.P. and A.M.; writing—review and editing, E.P., A.M., Y.M., P.T.N. and N.V.M.; visualization, E.P., Z.P. and T.M.; supervision, T.M.; project administration, E.P., A.M. and Y.M. All authors have read and agreed to the published version of the manuscript.

**Funding:** This research received no external funding.

**Informed Consent Statement:** Informed consent was obtained from all subjects involved in the study.

**Data Availability Statement:** The data presented in this study are available on request from the corresponding author. The data are not publicly available due to privacy restrictions.

**Acknowledgments:** The authors would like to thank all the participants who volunteered to participate in the study.

**Conflicts of Interest:** The authors declare no conflict of interest.

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
