# Peer review of "Changes in Cardiorespiratory Fitness during a Season in Elite Female Soccer, Basketball, and Handball Players"

_applsci, doi:10.3390/app13179593_

Round 1
Reviewer 1 Report
To the authors.
Thank you for submitting this paper for review. There are some interesting data in this study with some findings that may inform the training methods for female athletes in each of these sports. There are some key issues with the paper in its current form. The methods needs to fully described, including all details of the V̇O2max test, how blood pressure variables were measured, and more details regarding the ANOVAs calculated. The discussion also needs more detailed exploration of the findings and a deeper explanation as to what the data may mean and how it can be used by coaches and practitioners. Please see my detailed comments below, I look forward to reviewing the second submission of this work.
Kind regards,
Introduction
Please use the term ‘female’ throughout and avoid the term ‘women’ so the reader understands the authors are discussing sex rather than gender.
Please also include the dot over the V in ‘V̇O2max’.
Paragraph 1: Please provide data of the aerobic and anaerobic contribution to energy resynthesis in each of these sports to demonstrate the importance of aerobic capacity to performance.
Paragraph 2: Please expand on why there might be such a wide range of aerobic capacities in each of these populations.
Paragraphs 3 and 4: Please report the data from these studies showing how much aerobic capacity changes in these sports – are these changes positive or negative? Were these changes in response to a specific training program or their normal technical/tactical training? Which field tests tend to be used, and how do the results of these compare to the laboratory tests that have been reported?
Material and Methods
Study design
Paragraph 1: Please provide the ethics application reference number.
Paragraph 3: Please state which Performance Tier best describes the athletes using McKay et al. (2021) classification (https://journals.humankinetics.com/view/journals/ijspp/17/2/article-p317.xml). Please provide a brief breakdown of how many of the 5 training sessions per week were technical/tactical and how many were strength and conditioning sessions. As this study is reporting changes in fitness then it would be important to know how much fitness training the athletes completed.
Figure 1: This is a useful figure. Did all the athletes in each sport play in all games? How many sessions and games were missed due to injury?
Anthropometric measurements: Please use the terms ‘mass’ and ‘stature’ throughout in place of weight and height. Please state whether the skinfold measurements were taken by an ISAK Accredited practitioner or not.
Table 1: Are these differences between groups the post-hoc differences from an ANOVA? If so, was this a repeated measures ANOVA or a one-way ANOVA?
Cardiorespiratory endurance: Please provide more details regarding the protocol used. How was the intensity incremented? What speed did the athletes start at? How was the anaerobic threshold identified? How was V̇O2max identified? Why was lactate only measured 5 mins after recovery?
Results
Please calculate and report effects sizes for all ANOVAs to provide a more complete description of the findings and to allow inclusion in future meta-analyses.
Please also report the full ANOVA results, including the F and df values. Please also report the exact p values (unless p<.001).
Please make it clear which of the reported results are ANOVA comparisons and which are post hoc comparisons.
Table 2 and 3: When and how were SAP and DAP measured? These don’t seem to be mentioned in the methods section. Please label ‘VO2’ as ‘Absolute V̇O2max’, and please label ‘VO2max’ as ‘Relative V̇O2max’ to avoid any confusion for the readers.
Table 2 and Table 3 are showing the same data, just in different order and format. Table 2 is easier to read and understand as this shows the changes over the season within the sports, so please delete Table 3. Please include definitions for #, * and $ in the caption for Table 2.
Discussion
First paragraph: “VO2max increased in the middle of the season and had a slight drop at the end in all three sports.” – please rephrase, as the reduction in V̇O2max at the end of the season wasn’t ‘slight’. In all three sports it returned to at least baseline, with handball dropping below baseline. This indicates that any fitness improvements made during the season were lost by the end of the season. This is an important finding.
Second paragraph: “In VO2max the values were also higher in soccer players from the start of the preparation until the end of the season compared to the other two team sports.” – why might this be case? Why did this occur? What could this finding mean? Why might this be important? Do soccer players train differently or have different sport specific demands than athletes from the other two sports?
Fourth paragraph: What specific training types could have had these effects? How might these adaptations affect performance? How could the previously discussed fatigue be avoided? What training might be used to avoid the reduction in aerobic capacity reported in the final third of the season in each sport?
Fifth paragraph: It’s accepted that the microcycles were not recorded in detail, but an overview of what training types were used is important to understand the data.
Sixth paragraph: “The results of the study showed that cardiorespiratory performance improved significantly from baseline compared to mid-season measurement in all three sports.” Please make it clear that these all returned to baseline over the final third of the season.
Conclusion: “….as well as the probable change of motivation of the athletes who participated in the study.” How was motivation determined? This is why clearly stating how V̇O2max was confirmed is vital – if they met the criteria for V̇O2max in the test then motivation did not affect the results. If they only reached V̇O2peak then motivation might have been a problem. Please include this information in the methods (see previous comment) and adjust this conclusion accordingly.
English language quality is good considering the authors are not native English speakers. Some very minor sentence errors.
Reviewer 2 Report
To the authors:
The topic of the research is current, one of the main problems of competitive sport is that the training work of the period of preparation, competition and recovery is not as distinctly separated as before.
A particularly exciting task is the development of the abilities carried out throughout the year.
The thesis presents the changes in cardiorespiratory characteristics of 3 sports with 29-week follow-up studies.
From the point of view of women's sport, the results may be of interest.
Acknowledging the value of the research, I formulated the following problems and questions:
1. It is a common mistake that in this type of measurement, the first test is carried out immediately after the rest period. In the thesis there is no information about the period between the end of the competition period and the start of training work, the degree, the level of physical activity, the methods used. Improvement after a passive or inefficient period is evident and difficult to interpret. Regardless, this may be true.
2. Information used to improve the cardiorespiratory system (development, level keeping) is completely missing, load data (km) for 22 matches are known, but load zones are not.
3. As regards the methodology of the studies, the method of gas exchange tests is up-to-date, the procedure used to estimate body composition is acceptable. The measured and calculated variables discussed need to be supplemented.
4. The protocol used for stress tests is unknown. This is problematic, because, according to convention, about the 10-minute incremental load protocol seems suitable for measuring Vita-Maxima performance. The measured exercise times (7-8 minutes) assume that the work was terminated due to local fatigue, but this is contradicted by medium lactate values.
In addition to the BxB technique, what averaging was done on the sampling?
There could be no motivation problem, as the RER values were always above 1. The resulting image needs explanation. The discussion of oxygen uptake without load times is misleading and difficult to interpret.
Another problem is that the authors did not provide anthropometric data until the first measurement. If we estimate body weight by counting back from the absolute and relative oxygen uptake, it can be concluded that by the end of the competition period (3. Measurement) body weights are significantly reduced. This means that comparing the normalized values is problematic for the cardiorespiratory performance. It should be noted here that the load times, which represent real physical performance, practically did not change, there was a significant difference only in one case (basketball 7.73 - 7.30 minutes, which is not professionally interpretable)
Finally, sports comparisons are fair and valuable data.
The biggest problem in interpreting the results is the relationship between oxygen uptake and exercise time. What does it mean, if the exercise times do not change, but oxygen uptake does?
Unfortunately, the authors do not discuss the evaluation of the anaerobic threshold (AnT), the "shift right" characteristic is observable and should be interpreted.
In my opinion, it is possible to supplement the measured data (Oxygen pulse, Oxygen %, etc.) and take into the discussion.
Round 2
Reviewer 1 Report
To the authors,
thank you for responding to my review and requests in a timely and thorough manner. There are some minor changes still to make, after which I am happy to recommend publication. Thank you for your time, I look forward to seeing your work in print.
Kind regards
Throughout: Please be consistent in the annotation used for relative measures. For example, the authors have used ∙ in some instances (i.e., mmol∙L) and / in others (i.e., km/h). Please use one or the other throughout.
Methods
Cardiorespiratory measures: Thank you for adding the V̇O2max criteria. The authors do not seem to have collected lactate during the V̇O2max test, so please remove the use of lactate>6mmol∙L from this list of criteria as it could not have been used in this instance. Please also make it clear that the Borg 6-20 scale was used to avoid any potential confusion on the part of the readers.
Tables 4 and 5 (Tables 2 and 3 in old version): I agree that the tables are showing the data in different format (between time points vs. between sports), but they are still showing the same data so there is redundancy here. I’m happy for the authors to keep whichever table they feel show the data ‘best’ and delete the other one. I think either of the tables on its own allows the reader to compare both between sports and time points.
Reviewer 2 Report
The author's replies for the questions are accepted. Unfortunately, it is a problem with O2 uptake and running time remained open.
Author Response
Reply to reviewer 2
Dear reviewer,
as we mentioned in revisions 1, it is difficult to interpret the fact that they increased VO2max but did not increase exercise time. One factor that can affect is the running economy. It is well known that running economy can affect aerobic test performance. Athletes with the same VO2max show different performance due to different running economy (the energy required to cover a certain distance at a certain speed) female athletes may have increased VO2max but did not improve running economy, resulting in no differences in exercise time (running performance) [Bassett & Howley, 2000].
Do you want us to add something relevant to the text?
